# Revisiting the Role of Day 14 Bone Marrow Biopsy in Acute Myeloid Leukemia

**DOI:** 10.3390/cancers17050900

**Published:** 2025-03-06

**Authors:** Omer Jamy, Garrett Bourne, Todd William Mudd, Haley Thigpen, Ravi Bhatia

**Affiliations:** 1Division of Hematology and Oncology, Department of Medicine, University of Alabama at Birmingham, Birmingham, AL 35294, USA; 2Department of Internal Medicine, University of Alabama at Birmingham, Birmingham, AL 35294, USA

**Keywords:** AML, day 14 marrow, bone marrow biopsy, mid-induction marrow

## Abstract

Performing a day 14 bone marrow biopsy in acute myeloid leukemia is still a common practice, at least in the United States. The procedure is invasive, and its utility has become very questionable in recent years, with the main concern being the over-treatment of patients. In this review, we discuss published data highlighting the limitations of this procedure and future research aimed at omitting it from clinical practice.

## 1. Introduction

Acute myeloid leukemia (AML) is a heterogeneous hematological malignancy characterized by the rapid proliferation of myeloid progenitor cells in the bone marrow and peripheral blood that leads to significant impairment of normal hematopoiesis [1]. Over time, the estimated annual incidence of AML within the United States has increased from 3.4 to approximately 4.2 cases per 100,000 individuals indicating AML remains a significant public health concern [2]. While the 5-year overall survival rate of AML has remained at about 27%, it should be noted that prognosis varies widely according to age, cytogenetic abnormalities, and the presence of specific genetic mutations [3]. Most notable is the vast difference in survival based on patient age, with five-year overall survival (OS) in patients <60 years approaching between 40 and 50% vs. 5–10% in patients >60 years old. Survival also decreases significantly with accumulation of high-risk cytogenetic features and mutations [4,5,6].

Treatment of AML consists of an induction phase that typically consists of either intensive chemotherapy (IC) or lower-intensity chemotherapy (LIC). Criteria such as patient age, performance status, medical comorbidities, and disease biology factor into treatment decisions. AML is risk-stratified based on cytogenetic and mutational profiles as outlined in the updated 2022 European LeukemiaNet (ELN) recommendations that, among other outcomes, predicts relapse risk [7]. The goal of initial induction therapy is to obtain remission. Subsequently, based on risk stratification, consolidation may consist of either additional chemotherapy alone (favorable risk disease) or allogeneic stem cell transplantation (allo-SCT) (intermediate and adverse-risk disease) [1,6,7,8].

Intensive induction typically involves a combination of cytarabine and an anthracycline, generally termed the “7+3” regimen, with the addition of a third agent for certain subsets followed by consolidation with high-dose cytarabine (Hi-DAC). Studies have shown that the overall remission rates and long-term survival for those receiving IC are approximately 40–70% and 30–55%, respectively, with responses differing with age and disease risk classification [9,10,11,12,13]. Intensive induction is typically reserved for younger, fit patients with few comorbidities who can tolerate the regimen, whereas elderly and frail patients are better candidates for LIC. Recent advances in LIC regimens have provided more options for older patients. VIALE-A established the combination of hypomethylating agents and venetoclax (HMA/VEN) as the standard of care for LIC [14].

## 2. Bone Marrow Aspirate and Biopsy

Bone marrow aspirates and biopsies play a crucial role in the management of AML at multiple time points during the disease course. At diagnosis, a bone marrow biopsy is essential for confirming the presence of leukemic blast cells, assessing overall cellularity, and identifying cytogenetic and molecular abnormalities that inform prognosis and treatment strategies [7]. During intensive induction chemotherapy, a mid-induction bone marrow biopsy is typically performed around day 14 (D14) or day 21 (D21) (in the case of patients with FLT3-mutated AML receiving an FLT3 inhibitor) as an initial assessment of response to treatment [15,16]. The morphologic evaluation of the mid-induction biopsy is, although, very simple and inexpensive to perform, but, at the same time, can be very non-specific. Patients with persistent disease (presence of >5–10% blasts) and a hypercellular bone marrow will typically undergo reinduction chemotherapy, although some of these patients might achieve complete remission (CR) without immediate reinduction. Those with hypoplastic (<20% cellularity and <5% residual blasts) bone marrow will usually be monitored for count recovery. The decision to proceed with immediate reinduction has generally been at the discretion of the treating physician, with no clear guidelines. After the completion of induction chemotherapy, follow-up bone marrow evaluations are crucial for assessing response to induction chemotherapy, monitoring for disease relapse, and planning for consolidation chemotherapy and/or allo-SCT.

## 3. Mid-Induction Bone Marrow Analysis

In recent years, the practice of routinely obtaining D14 bone marrow biopsies during AML induction therapy has been scrutinized. While current National Comprehensive Cancer Network (NCCN) guidelines recommend obtaining D14 biopsies to gauge early response to treatment and guide potential changes in future management, concerns have been raised that D14 biopsies may not be as prognostically accurate as hoped and subsequently may result in additional and unwarranted chemotherapy in select patients [15,16,17]. In this review, we aim to comprehensively summarize and synthesize both past and more recent evidence surrounding D14 or D21 bone marrow biopsies along, with ongoing clinical trials in this space, outlining what future goals should be. A summary of the studies discussed is provided in Table 1.

Before assessing the specific utility of D14 bone marrow biopsies during induction therapy, it is important to first understand the various strengths and limitations of bone marrow biopsies in general. Bone marrow biopsies involve multiple different components that provide pathological and diagnostic information. For the scope of this review, we will focus on the utility of these components in a D14 bone marrow evaluation. A bone marrow biopsy also allows for cytogenetic and molecular analysis in addition to microscopic evaluation of the cells and architecture of the marrow. These tests can be repeated when D14 bone marrow biopsies are performed. The exact sensitivity and specificity of cytogenetic and molecular testing in detecting AML varies based on the targets and markers used due to the heterogeneous nature of these tests. Studies have shown that immunotyping of cells using flow cytometry for analysis and classification based on cell surface markers have sensitivity and specificity values of 94.4% and 99.9%, respectively [32,33]. Cytogenetic analysis of chromosomes for karyotype and specific chromosomal translocations is also highly sensitive and specific, but the exact figures are dependent on the assays and targets used [34]. Analysis of the molecular mutational profile in AML has become very effective, with sensitivity thresholds approaching 10^−5^–10^−6^, though exact sensitivities differ based on the testing used and sample quality [35,36]. Notably, molecular mutation profiles have been shown to be equally effective when performed on peripheral blood or bone marrow samples and can be used for disease surveillance even when no residual disease is present [37].

When evaluating the utility of bone marrow biopsies, it is also important to acknowledge inter-pathologist variation. This variability can affect the accuracy and consistency of diagnoses, which is critical for determining appropriate treatment pathways. A study by Tan and Lamberg highlighted the variability in morphologic criteria used to diagnose acute leukemia, emphasizing that different Romanowsky stains can lead to marked differences in the appearance of blast cells, except for the nuclear-cytoplasmic ratio [38]. This suggests that even within a single institution, the choice of staining technique can introduce variability in pathologic assessments. Furthermore, Matsuda et al. investigated interobserver concordance in the assessment of dysplasia and blast counts, finding that while there was generally moderate to high agreement, discrepancies still existed, particularly in the assessment of dysmegakaryopoiesis [39]. These findings indicate that different pathologists may arrive at different conclusions based on the same biopsy material, emphasizing the need for standardized protocols and continuous quality assessments to minimize subjectivity and diagnostic discrepancies.

Though they acknowledge there is limited prospective data to support the recommendation, the 2024 NCCN AML guidelines still include instructions to consider a follow-up bone marrow aspirate and biopsy 14–21 days after the start of therapy with reinduction for those with residual disease or without hypocellular marrow [17]. Specific criteria for marrow response vary by institution, but typically <5% blasts and <20% cellularity is considered adequate to defer reinduction. Despite this, there remains considerable concern that a significant number of patients undergo unnecessary chemotherapy because early post-induction biopsies do not always accurately predict response.

## 4. Can Day 14 Biopsy Results Predict Remission?

The ability to predict which patients are going to achieve CR after single induction therapy is of notable clinical importance. If a patient can be accurately predicted to achieve CR, they do not need to be exposed to the toxicity of double induction therapy. Meanwhile, it has been theorized that if a patient can be accurately predicted to not achieve CR after initial induction, then they should not experience delays in starting additional therapy. However, whether reinduction needs to be administered immediately or can be delayed by 1 or 2 weeks is not clear. Recent data in the upfront setting suggests that a treatment delay from the time of diagnosis does not have a negative impact on prognosis [40].

## 5. Sensitivity of Day 14 Biopsy

Prospective data suggesting the accuracy in predicting a patient’s response after induction using bone marrow analysis has largely been limited to European clinical trials where double-induction was preplanned, but there have been several retrospective studies that report similar findings. Most studies to date utilize leukemic blast cell percentage as their major prognostic indicator. These studies note a correlation between lower leukemic blast cell percentages and higher rates of CR, but the exact cutoff varies between studies [18,19,20,21,22,23,24,25]. Liso et al. were among the first to demonstrate that day 14 biopsies may be predictive of treatment response, with the caveat that patient-specific prognostic factors changed the overall likelihood of predicting remission based on a finite blast cell cutoff. They examined the relationship between higher D14 blast cell cutoffs and achieving CR, but in doing so, they noted that different blast cell cutoffs were associated with higher prognostic accuracy in different age groups. For reference, patients < 60 years old had a complete response rate of 79%, with D14 leukemic blasts < 22% (sensitivity, 93.9%), but patients > 60 years old had a complete response rate of 67% (sensitivity, 66.7%) with D14 leukemic blasts <15% [18]. In regard to finding an ideal blast cell cutoff, previous studies have noted lower rates of CR with <10% blast cells on D14 bone marrow when compared to studies using a blast % of <5% [19,20,21,23,25]. The sensitivity of the D14 bone marrow biopsy increases as we lower the blast percentage. This is further emphasized by Kern et al., who showed that, as a continuous variable, blast cell percentage was related to CR [19]. Many of the aforementioned studies found that D14 bone marrows were highly sensitive in regard to recognizing which patients would achieve CR. Despite this, there remains a ceiling to the sensitivity of detecting future responders with a <5% blast cutoff, as many of these studies noted an approximately 5–10% failure rate to identify patients who would not achieve remission despite having low blasts on the D14 bone marrow [16].

Othus et al. noted that despite cooperative group protocols typically calling for a second induction if the first was unsuccessful by D14 biopsy, only about 52% of patients enrolled in four different SWOG protocols received this follow-up therapy [41]. This notable deviation from protocol aligns with other literature suggesting there is uncertainty about the management of patients not in CR after initial induction. In the next section, we focus on the specificity of the D14 bone marrow biopsy.

## 6. Specificity of D14 Biopsy

Even among the literature arguing against the routine use of D14 biopsies, it is agreed that <5% blasts on D14 biopsy are highly predictive of achieving CR at the end-of-induction. However, multiple studies have noted that the specificity of D14 biopsies with ≥5% blasts to predict failure to achieve CR is poor [15,26,27,28,30,31,42]. Liso et al. demonstrated that for patients < 60 years, using a D14 blast cutoff of >22% resulted in 19% of patients achieving CR without further chemotherapy (specificity 71.4%), and using a D14 blast cutoff of >15% resulted in 19% patients achieving CR without further chemotherapy (specificity 80.9%) in patients ≥ 60 years. Hussein et al. investigated whether cytoreduction at D14 correlated with a complete response on D28 using a blast cell cutoff <5%. They found that cytoreduction at day 14 positively predicted CR (PPV 79%,) but that a lack thereof did not indicate that remission would not be achieved (NPV 29%) [18]. Jamy et al. looked specifically at whether double induction changed outcomes for patients with indeterminate D14 biopsies in two similar cohorts and found that the rates of CR in those who received additional induction therapy and those who did not were similar [15]. This reemphasized the work conducted by Morris et al. and Bertoli et al. that showed that of the patients with >5% blasts on mid-induction marrows, many would go on to achieve CR without further induction therapy [21,27]. The concern raised by many of these authors is that a nontrivial number of patients with ≥5% blasts on D14 biopsy will ultimately achieve CR without further intervention. In fact, Yezefski et al. estimated that routine D14 biopsy could result in up to 76% of patients with inadequate D14 biopsy responses receiving additional, unnecessary induction chemotherapy [29]. The decision to give an immediate reinduction or not can be challenging. The purpose of therapy is to achieve deep remission, but the benefit or reinduction needs to be weighed against the potential risks of intensive chemotherapy. Given that nearly 50% of patients with >5% blasts on mid-induction bone marrow biopsy may receive unnecessary chemotherapy, it is essential to refine this clinical practice.

## 7. Pooled Sensitivity and Specificity Analysis

From the studies described above, the mean pooled sensitivity for a mid-induction biopsy result of <5% blasts to predict remission at the end of induction was 91.5% (95% CI: 88.1–95.3%). The mean pooled specificity for a mid-induction biopsy result of ≥5% blasts to predict failure to achieve remission at the end of induction was 53.6% (95% CI: 43.1–61.4%). Further analysis suggested that with the assumption of refractory leukemia to one cycle of induction of 10% and 20%, the positive predictive value of mid-induction biopsy was 29% and 47%, respectively. The methodology utilized for this analysis was published by Reitsma et al. [43].

## 8. Alternative Options to Day 14 Biopsies

As previously outlined, significant hesitation regarding the routine utilization of D14 bone marrow biopsies to inform decision-making on second induction remains despite NCCN guidelines recommending the practice. Considering these reservations, several alternative strategies have emerged to assess early treatment response, which we will explore in this section.

One alternative to the D14 biopsy is to perform the same procedure but at a different time point. Ofran et al. performed a biopsy on day 5, in addition to day 14 biopsies in 127 patients prospectively. The results of day 5 marrow were observational. On day 5 and day 14, 34.3% and 61.2% of patients had <10% blasts, respectively. These patients were identified as early responders and had similar CR rates. By moving the biopsy time point earlier in the treatment course, the specificity of day 5 was high at 88%, but this came at the expense of a significantly decreased sensitivity of 45.6%, highlighting the failure of day 5 biopsy to identify slow responders [44]. Conversely, Yanada et al. evaluated the role of day 21 bone marrow biopsy in 586 patients who also had a D14 marrow performed. They reported that out of 72 patients with 20–59% blasts on day 14, 37/72 (51%) of those patients had <20% blasts on day 21, and ultimately 23/37 (62%) would go on to achieve CR without further chemotherapy [26]. They further reported that the results of a day 21 bone marrow did not change the decision to administer reinduction, regardless of the disease burden on day 14 bone marrow biopsy. Currently, the practice of performing a day 21 bone marrow biopsy is restricted to those patients receiving an FLT3-inhibitor, mainly due to the drug being administered on days 8–21 of induction.

Another suggestion to enhance the utility of the D14 biopsy is the integration of predictive models into clinical decision-making. Norkin et al. developed a predictive model that incorporated both clinical and laboratory parameters to the results of the D14 marrow to improve the accuracy of predicting treatment outcomes. In 89 patients with residual disease on the D14 biopsy, 39 went on to achieve CR without further chemotherapy. In addition to the D14 bone marrow blast %, factors that predicted CR included relapse disease, adverse-risk disease, and treatment-related and secondary AML. This study suggested that the addition of clinical and laboratory markers to calculate a patient’s overall risk increases specificity (93%) in predicting non-responders, but may fail to identify patients who will eventually respond to induction (sensitivity 78%) [30]. Furthermore, Eckardt et al. employed supervised machine learning techniques utilizing various laboratory parameters to predict complete remission and overall survival [45]. Together, these approaches highlight the potential of advanced analytics in refining treatment strategies and suggest a paradigm shift toward more personalized and data-driven decision-making.

Another alternative monitoring strategy we discuss is the rate and degree of peripheral blast cell decline. Lacombe et al. demonstrated that multiparametric flow cytometry allows for the highly sensitive detection of blast cells and noted that the time required to reach 90% peripheral blast depletion was highly predictive of CR [46]. These findings are supported by the work of Gao et al., who noted that the percentage of peripheral blast cells on day 7 post-induction is a significant predictor of both complete remission and overall survival [47]. Arellano et al. further supported this notion by showing that peripheral blast clearance by day 14 post-induction is also a strong indicator of both CR and OS [48]. Collectively, these findings underscore the importance of peripheral blood monitoring as a critical component in assessing treatment response, potentially offering a more accessible and less invasive alternative to routine bone marrow biopsies. However, not all patients present with blasts in the peripheral blood and, therefore, this strategy may not be applicable to all patients undergoing induction. However, where possible, the rate of peripheral blood clearance of blasts should be used in conjunction with the results of the D14 bone marrow biopsy to improve accuracy.

More recently, the mid-induction marrow cellularity has been shown to impact the outcomes of induction therapy. Griffin et al. evaluated 176 patients who received intensive reinduction for residual disease (defined as >10% blasts) on D14 bone marrow biopsy. The majority of the patients had intermediate (58.5%) and adverse-risk (38.1%) cytogenetics. The rate of response with a second induction was 59%. In multivariable analysis, young age, de novo disease, and non-adverse karyotype were predictive of remission and survival. They found that a hypocellular marrow at D14 improved the likelihood of CR after reinduction (72.4% vs. 42.6%,) albeit it is unclear if these patients would have gone on to achieve CR without a second induction. In multivariable analysis, a hypocellular marrow was predictive of both remission and survival. Interestingly, an absolute blast count reduction >50% was significant in univariate analysis but not in multivariable analysis [22]. Ollila et al. also demonstrated that a hypocellular marrow resulted in a higher remission and survival rate in a study of 31 patients. Although the sample size was small, they also found no difference between reinduction and outcomes and demonstrated better specificity with a higher blast % and marrow cellularity [42]. A study by Jamy et al. went on to investigate this question, in more detail, by focusing on patients with an indeterminate D14 bone marrow result defined as <20% cellularity and between 5 and 20% blasts. They analyzed 50 such cases, of which 25 received immediate reinduction and 25 were observed. The rate of CR between both groups was similar, as were the long-term survival outcomes, again questioning the utility of immediate reinduction and raising the concern for overtreatment in this patient population [15]. These three studies indicate that marrow cellularity has important prognostic implications, though it is still unclear if subjecting patients to additional chemotherapy will change their overall outcomes.

## 9. Genomics and Day 14 Biopsy

The remission rate with ‘7+3’ ranges from 40 to 70%, depending on several patients and disease characteristics. Among disease factors, risk stratification based on cytogenetics and molecular aberrations plays a vital role in determining outcomes. Jamy et al. reported no difference in remission rate or survival between observation vs. immediate reinduction for patients with residual disease on D14 bone marrow. In their study, both groups had approximately 50% of patients with adverse-risk diseases based on either cytogenetics or molecular analysis. Even after excluding patients with favorable-risk disease, the remission rates between the two groups remained the same [15]. England et al. evaluated 486 patients with the D14 evaluation and reported that cytogenetic risk and D14 blasts < 5% were predictive of both remission rate and survival. Interestingly, they found that a second induction given for residual disease on the D14 bone marrow did not lead to a survival advantage, highlighting that overall prognosis is impacted by risk stratification [23]. We previously reported a remission rate of only 10.3% for biallelic TP53-mutated AML treated with ‘7+3’. Nearly all patients had significant residual disease on D14 bone marrow biopsy, and treatment was switched to either high-dose cytarabine-based therapy or the hypomethylating agent +/− venetoclax (HMA+VEN) [49]. We now know the aggressive biology of biallelic TP53-AML and recommend against induction with ‘7+3’ given the dismal outcomes and, therefore, alleviate the need for a mid-induction marrow. More recently, in the QuANTUM—first trial, a second induction was administered for ≥5% blasts on either day 21 or day 28 bone marrow. The study enrolled 539 patients, and only 20% of them received a second induction. Details of the burden of the disease for which a second induction was administered, as well as the timing of the second induction (after day 21 or after day 28), are not available to analyze the added benefit of a second induction in FLT3-mutated AML. Nonetheless, the CR rate was similar in both arms (quizartinib vs. placebo) [11]. To conclude, genomics at diagnosis are well-recognized independent predictors of outcome and should be utilized as such. How this piece of information guides management in the context of the kinetics of the mid-induction marrow is not entirely clear. What is clear is that cytogenetic and molecular data are very helpful in predicting the end of induction response and should guide upfront treatment selection for patients.

## 10. Day 14 Biopsy with Other AML Therapies

Most of the data available today for mid-induction bone marrow biopsy are in the context of ‘7+3” therapy. In the past 5–10 years, we have seen several new drugs approved for the treatment of both newly diagnosed or relapse/refractory AML. Here, we discuss the mid-induction bone marrow biopsy data for the drugs approved for newly diagnosed AML, acknowledging that is challenging to extrapolate the data from the ‘7+3’ population to this setting.

### 10.1. CPX-351

CPX-351, a liposomal formulation of cytarabine and daunorubicin, was approved for the treatment of patients with newly diagnosed secondary AML based on a survival advantage compared to ‘7+3’ in a randomized trial [50]. The trial focused on patients 60–75 years with high-risk AML, defined as either therapy-related, AML with antecedent MDS or CMML or AML with myelodysplasia-related changes. The study administered a second induction for patients not achieving hypoplastic D14 marrow. CPX-351 resulted in a higher CR rate compared to ‘7+3’ (37.3% vs. 25.6%, *p* = 0.04). The composite remission rate was also higher with CPX-351 (55.2% vs. 34.0%) after one induction cycle. Interestingly, among patients with two induction cycles, there was no difference in response rate (CPX-351 = 31.3%, 7+3 = 35.3%), despite CPX-351 leading to a survival benefit for the entire population. Granular details of disease burden and timing of the second induction were not provided to further analyze mid-induction biopsy results. Furthermore, the median time to neutrophil recovery was prolonged with CPX-351 with either one or two induction cycles (35 days vs. 29 days). Given similar response rates to second induction as well as prolonged time to count recovery, the utility of a D14 biopsy in recipients of CPX-351 remains unclear, and further studies are needed to understand it better.

### 10.2. HMA+VEN

The combination of HMA+VEN is now considered the standard of care for older or unfit patients with newly diagnosed AML. The initial study utilized 28 days of VEN [14]. As HMA+VEN became routinely adapted in clinical practice, several modifications to the initial treatment schedule have been recommended to minimize the myelosuppressive toxicities of the regimen. One recommendation is to perform a biopsy between days 21 and 28, and to hold VEN if a morphological leukemia-free state is achieved, as a measure to assist in count recovery. Several studies are now being conducted to optimize the dose and duration of VEN during induction therapy [51]. More recently, the addition of a third agent to HMA+VEN (triplet therapy) is being investigated in clinical trials. Although these are not yet considered standard, the US Food and Drug Administration (FDA) has provided guidance in designing trials to manage the toxicities of the combination. Specifically, in the setting of triple therapy with HMA+VEN, a day 14 bone marrow biopsy is recommended and in the setting of a morphological leukemia-free state, VEN +/− the investigational agent can be held to assist in count recovery [52,53]. Most of the mid-induction biopsy data in the setting of HMA+VEN is to guide holding further therapy instead of immediate reinduction. In the setting of disease on the biopsies performed in recipients of HMA+VEN, the recommendation is to continue VEN per the prescribing insert.

## 11. Limitation of Current Data

Based on the data discussed, the mid-induction bone marrow biopsy is sensitive in predicting the ability to achieve CR, but even that sensitivity has a ceiling. At the same time, there is no specificity in predicting the failure to achieve remission. Several attempts have been made to increase the specificity of this procedure, but that comes with the cost of decreased sensitivity. There is enough data to suggest that up to 50% of patients with ‘residual disease’ on a mid-induction bone marrow biopsy may go on to achieve remission without immediate reinduction chemotherapy, raising the concern that we are administering unnecessary chemotherapy to some of these patients. These studies also raise the question of whether residual disease on a mid-induction marrow is a marker of poor disease biology. Furthermore, while several studies discussed above noted equivalent outcomes between those who received reinduction, and those who were only observed, the retrospective nature of most of these studies makes it difficult to assess if patients who were reinduced would have experienced different outcomes if they had not received further therapy. Nonetheless, the mid-induction marrow biopsy is an invasive procedure, and its utility has become very questionable.

## 12. Conclusions

Performing a D14 bone marrow biopsy is still a common practice, at least in the US, and is endorsed by the NCCN guidelines. Part of the reason is that the quality of evidence arguing for eliminating the procedure is largely retrospective. Nonetheless, it is worth revisiting the role of the mid-induction bone marrow biopsy in clinical practice. Strategies to overcome the subjectivity of the procedure may include the application of sensitive tests such as flow cytometry, next-generation sequencing (NGS), or polymerase chain reaction (PCR) to the marrow sample, but this may be limited based on a hypoplastic marrow. Trials are also investigating the role of positron emission tomography (PET) scans, in AML, to help assess early response to treatment (NCT02392429). Prospective validation of the findings of the studies discussed in this review will assist in limiting and perhaps even eliminating the D14 bone marrow procedure. Ideally, a randomized trial of either treatment vs. no treatment for those with residual disease on D14 or performing D14 vs. omitting it is needed to assess the benefit of this procedure. Unfortunately, such a trial is challenging to perform given the sample size needed and even a lack of interest due to being a non-therapeutic trial. At our institution, we are currently prospectively evaluating the utility of the mid-induction bone marrow biopsy in AML (NCT06323590). Our design includes performing a D14 or D21 marrow on all patients receiving ‘7+3’ induction chemotherapy and observing till the end-of-induction marrow, regardless of the results of the mid-induction marrow. Additionally, we are performing RNA sequencing on the diagnostic, mid-induction, and end-of-induction marrow samples to characterize the kinetics as well as the nature of the cells in the bone marrow through induction. By not intervening in the marrow results in a controlled environment on nearly consecutive patients, we hope that our results will help refine the role of mid-induction bone marrow biopsy in AML.

To conclude, data from numerous studies lay the framework for larger prospective studies to aid in omitting mid-induction bone marrow biopsy from routine practice and instead provide guidance regarding further therapy based on the results of the end-of-induction bone marrow biopsy.

## Figures and Tables

**Table 1 cancers-17-00900-t001:** Key findings of studies evaluating mid-induction bone marrow biopsies.

Study	N	Key Findings
Liso et al. (Retrospective) Ref. [18]	198	Risk stratification: Unavailable Treatment: 7+3, ICE, MEC Patients < 60 y: ≤22% blasts resulted in 79% achieving CR while >22% blasts resulted in 81% non-responders (*p* < 0.0001) Patients ≥ 60 y: Blast percentage ≤ 15% blasts resulted in 67% achieving CR while >15% blasts resulted in 81% non-responders (*p* = 0.0001)
Kern et al. (Prospective) Ref. [19]	449	Risk stratification: Fav CG = 10%, Int CG = 48.3%, Adv CG = 28.5% Treatment: TAD, HAM <10% blasts resulted in 83.75% CR while ≥10% blasts resulted in 53.61% CR (*p* < 0.0001)
Hussein et al. (Retrospective) Ref. [20]	194	Risk stratification: Fav CG = 5.6%, Int CG = 27.3%, Adv CG = 31.4% Treatment: 7+3 +/− etoposide ≤5% blasts was strongly predictive of CR with 90% sensitivity and 79% positive predictive value
Bertoli et al. (Prospective) Ref. [21]	823	Risk stratification: Fav CG = 15%, Int CG = 57.6%, Adv CG = 20.8% Treatment: 7+3 (or 5 days of idarubicin) <5% blasts resulted in 91.7% CR while ≥5% blasts resulted in 69.2% CR (*p* < 0.0001)
Griffin et al. (Retrospective) Ref. [22]	176	Risk stratification: Fav CG = 1.1%, Int CG = 58.5%, Adv CG = 38.1% Treatment: 7+3 ≥50% blast reduction resulted in 68.4% CR/CRi while <50% blast reduction resulted in 48.6% CR/Cri (*p* = 0.03) Hypocellular marrow resulted in 72.4% CR/CRi while hypercellular marrow resulted in 42.6% CR/CRi (*p* < 0.001)
England et al. (Retrospective) Ref. [23]	486	Risk stratification: Fav CG = 13.8%, Int CG = 65.8%, Adv CG = 16.5% Treatment: 7+3 <5% blasts resulted in 87% CR/CRi while ≥5% blasts resulted in 56% CR/CRi (*p* < 0.001)
Manuprasad et al. (Retrospective) Ref. [24]	96	Risk stratification: Fav CG = 17%, Int CG = 66%, Adv CG = 17% Treatment: 7+3 <5% blasts resulted in 98% CR while ≥5% blasts resulted in 88% CR (*p* = 0.04)
Lemos et al. (Retrospective) Ref. [25]	374	Risk stratification: Unavailable Treatment: 7+3 <10% blasts resulted in 62% CR while ≥10% blasts resulted in 23% CR (*p* < 0.001)
Yanada et al. (Retrospective) Ref [26]	586	Risk stratification: Fav CG = 6%, Int CG = 61%, Adv CG = 33% Treatment: High-dose cytarabine + idarubicin +/− fludarabine/topotecan 37 of the 72 patients (51%) with 20–59% blasts on D14 had <20% blasts on day 21, and 23 of the 37 (62%) entered CR without further therapy
Morris et al. (Retrospective) Ref. [27]	74	Risk stratification: Fav CG = 2.7%, Int CG = 50%, Adv CG = 20.3% Treatment: 7+3 11/13 patients with suboptimal D14 response who were observed without therapy attained a morphologic CR
Deutsch et al. (Retrospective) Ref. [28]	98	Risk stratification: Unavailable Treatment: Unavailable Achieving an optimal response at D14 was predictive of achieving CR at recovery (sensitivity = 95.3%, PPV = 97.6%). However, not achieving an optimal response at D14 had low specificity (50%) and NPV (33.3%) for achieving CR (*p* = 0.021)
Yezefski et al. (Retrospective) Ref. [29]	154	Risk stratification: Fav CG = 19.4%, Int CG/Adv CG = 80.6% Treatment: 7+3, high dose cytarabine 21 of the 26 patients never received reinduction despite having >5% blasts in the D14 marrow, but 16 of the 21 entered CR
Norkin et al. (Retrospective) Ref. [30]	183	Risk stratification: Fav CG/Int CG = 35.9%, Adv CG = 64.1% Treatment: 7+3 Of the 89 patients with leukemia-positive D14 biopsy who did not receive reinduction, 32 (36%) subsequently achieved CR/CRi
Alsaleh et al. (Retrospective) Ref. [31]	84	Risk stratification: Fav CG/Int CG = 38%, Adv CG = 35% Treatment: 7+3 Failure to show complete remission on D14 biopsy had only 60% specificity in predicting the failure of CR on D28 biopsy
Jamy et al. (Retrospective) Ref. [15]	50	Risk stratification: Fav CG/Int CG = 52%, Adv CG = 48% Treatment: 7+3 In patients with indeterminate D14 biopsies, CR/CRi rates were similar between patients treated with reinduction vs. observation

CR: complete remission, CRi: CR with incomplete hematological recovery; D14: day 14, PPV: positive predictive value, NPV: negative predictive value; D28: day 28, ICE: idarubicin, cytarabine, and etoposide; MEC: mitoxantrone, etoposide, cytarabine; TAD: thioguanine, cytarabine, and daunorubicin; HAM: cytarabine and mitoxantrone; CG: cytogenetics; Fav: favorable, Int: intermediate, Adv: adverse.

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
