# Peer review of "Revisiting the Role of Day 14 Bone Marrow Biopsy in Acute Myeloid Leukemia"

_cancers, 2025, doi:10.3390/cancers17050900_

Round 1
Reviewer 1 Report
Comments and Suggestions for Authors
This study summarized previously published data and concluded the significance of Day 14 bone marrow (BM) examination during induction chemotherapy in patients with AML.
The authors presented the previously published data in a table but did not conduct a systematic review. Therefore, I recommend performing a meta-analysis to determine the statistical significance of Day 14 BM.
Additionally, the prognosis and outcomes of AML are heterogeneous, influenced by genetic rearrangements and the presence of FLT3-ITD mutations, among other factors. Thus, the prognostic significance of Day 14 BM should be analyzed based on risk groups.
Author Response
Reviewer 1
This study summarized previously published data and concluded the significance of Day 14 bone marrow (BM) examination during induction chemotherapy in patients with AML.
The authors presented the previously published data in a table but did not conduct a systematic review. Therefore, I recommend performing a meta-analysis to determine the statistical significance of Day 14 BM.
We thank the reviewer for their comment. We have now analyzed all the studies and provided the mean pooled analysis for sensitivity and specificity. From the studies described above, the mean pooled sensitivity for a mid-induction biopsy result of <5% blasts to predict remission at the end of induction was 91.5 % (95% CI: 88.1%-95.3%). The mean pooled specificity for a mid-induction biopsy result of ≥5% blasts to predict failure to achieve remission at the end of induction was 53.6% (95% CI: 43.1%-61.4%). Further analysis suggested that with the assumption of refractory leukemia to once cycle of induction of 10% and 20%, the positive predictive value of mid-induction biopsy was 29% and 47%, respectively. The methodology utilized for this analysis was published by Reitsma et al. This information has been added to pages 5-6.
We would like to highlight that the purpose of our invited article was to provide a comprehensive review regarding the published literature regarding Day 14 bone marrow biopsy in AML along with future directions. We did not intend to conduct a meta-analysis but have provided results using appropriate methodology, nonetheless.
Additionally, the prognosis and outcomes of AML are heterogeneous, influenced by genetic rearrangements and the presence of FLT3-ITD mutations, among other factors. Thus, the prognostic significance of Day 14 BM should be analyzed based on risk groups.
We thank the reviewer for their comment. We have now added section 9: Genomics and Day 14 Biopsy on page 7. In this section we discuss the following:
- Genomics and Day 14 Biopsy
The remission rate with ‘7+3’ ranges from 40%-70%, depending on several patients and disease characteristics. Among disease factors, risk stratification based on cytogenetics and molecular aberrations play a vital role in determining outcomes. Jamy et al. reported no difference in remission rate or survival between observation vs. immediate re-induction for patients with residual disease on D14 bone marrow. In their study, both groups had approximately 50% patients with adverse-risk disease based on either cytogenetics or molecular analysis. Even after excluding patients with favorable-risk disease, the remission rates between the two groups remained the same. England et al. evaluated 486 patients with D14 evaluation and reported that cytogenetic risk and D14 blasts <5% were predictive of both remission rate and survival. Interestingly, they found that a second induction given for residual disease on the D14 bone marrow did not lead to a survival advantage, highlighting that overall prognosis is impacted by risk stratification. We previously reported a remission rate of only 10.3% for biallelic TP53-mutated AML treated with ‘7+3’. Nearly all patients had significant residual disease on D14 bone marrow biopsy and treatment was switched to either high-dose cytarabine based therapy or hypomethylating agent +/- venetoclax (HMA+VEN). We now know the aggressive biology of biallelic TP53-AML and recommend against induction with ‘7+3’ given the dismal outcomes and therefore alleviating the need for a mid-induction marrow. More recently, in the QuANTUM-First trial, a second induction was administered for ≥5% blasts on either a Day 21 or Day 28 bone marrow. The study enrolled 539 patients and only 20% received a second induction. Details of the burden of disease for which a second induction was administered as well as timing of second induction (after day 21 or after day 28) are not available to analyze the added benefit of a second induction in FLT3-mutated AML. Nonetheless, the CR rate was similar in both arms (quizartinib vs. placebo). To conclude, genomics at diagnosis are well recognized independent predictors of outcome and should be utilized as such. How this piece of information guides management in the context of the kinetics of the mid-induction marrow is not entirely clear. What is clear is that cytogenetic and molecular data are very helpful in predicting end of induction response and should guide upfront treatment selection for patients.
Reviewer 2 Report
Comments and Suggestions for Authors
The authors review the necessity of bone marrow examination on day 14 of remission induction therapy for AML, and introduce that current data raise questions about the necessity of bone marrow examination on day 14, and that they are conducting a clinical trial to clarify this.
As the author also states, since the responsiveness to chemotherapy differs depending on chromosomal abnormalities and gene abnormalities in AML, I believe that the judgment based on residual disease in the bone marrow on day 14 should also differ, but such a discussion on the judgment based on the AML subtype is not included in the text, so I would like you to add it.
Recently, remission induction therapy with new drugs such as CPX-351 has been implemented, and I think that the significance of bone marrow examination on day 14 differs depending on the treatment performed. I think it would be useful in current AML treatment if the text also included a further review of the significance for each treatment method.
In VEN/AZA therapy, it is proposed to suspend venetoclax in order to avoid prolongation of neutropenia if a bone marrow examination is performed in the middle and sufficient reduction of leukemia cells is obtained. I would like to hear your opinion on this.
Since Table 1 does not include references, please add them. Also, please add information on how many patients with what cytogenetic risk were included, what regimen was used for treatment, and whether it was a prospective or retrospective study to Table 1.
Since there are multiple clinical trial protocols for treatment development that stipulate that re-induction therapy should be performed in the intermediate bone marrow examination, I think that most studies cannot be said to be retrospective studies. Please distinguish between developmental studies and retrospective studies.
The same content is repeated throughout the text, so please organize the text.
References 44 and 45 are duplicated, so please correct them.
Author Response
Reviewer 2
The authors review the necessity of bone marrow examination on day 14 of remission induction therapy for AML, and introduce that current data raise questions about the necessity of bone marrow examination on day 14, and that they are conducting a clinical trial to clarify this.
As the author also states, since the responsiveness to chemotherapy differs depending on chromosomal abnormalities and gene abnormalities in AML, I believe that the judgment based on residual disease in the bone marrow on day 14 should also differ, but such a discussion on the judgment based on the AML subtype is not included in the text, so I would like you to add it.
We thank the reviewer for their comment. We have now added section 9: Genomics and Day 14 Biopsy on pages 7-8. In this section we discuss the following:
- Genomics and Day 14 Biopsy
The remission rate with ‘7+3’ ranges from 40%-70%, depending on several patients and disease characteristics. Among disease factors, risk stratification based on cytogenetics and molecular aberrations play a vital role in determining outcomes. Jamy et al. reported no difference in remission rate or survival between observation vs. immediate re-induction for patients with residual disease on D14 bone marrow. In their study, both groups had approximately 50% patients with adverse-risk disease based on either cytogenetics or molecular analysis. Even after excluding patients with favorable-risk disease, the remission rates between the two groups remained the same. England et al. evaluated 486 patients with D14 evaluation and reported that cytogenetic risk and D14 blasts <5% were predictive of both remission rate and survival. Interestingly, they found that a second induction given for residual disease on the D14 bone marrow did not lead to a survival advantage, highlighting that overall prognosis is impacted by risk stratification. We previously reported a remission rate of only 10.3% for biallelic TP53-mutated AML treated with ‘7+3’. Nearly all patients had significant residual disease on D14 bone marrow biopsy and treatment was switched to either high-dose cytarabine based therapy or hypomethylating agent +/- venetoclax (HMA+VEN). We now know the aggressive biology of biallelic TP53-AML and recommend against induction with ‘7+3’ given the dismal outcomes and therefore alleviating the need for a mid-induction marrow. More recently, in the QuANTUM-First trial, a second induction was administered for ≥5% blasts on either a Day 21 or Day 28 bone marrow. The study enrolled 539 patients and only 20% received a second induction. Details of the burden of disease for which a second induction was administered as well as timing of second induction (after day 21 or after day 28) are not available to analyze the added benefit of a second induction in FLT3-mutated AML. Nonetheless, the CR rate was similar in both arms (quizartinib vs. placebo). To conclude, genomics at diagnosis are well recognized independent predictors of outcome and should be utilized as such. How this piece of information guides management in the context of the kinetics of the mid-induction marrow is not entirely clear. What is clear is that cytogenetic and molecular data are very helpful in predicting end of induction response and should guide upfront treatment selection for patients.
Recently, remission induction therapy with new drugs such as CPX-351 has been implemented, and I think that the significance of bone marrow examination on day 14 differs depending on the treatment performed. I think it would be useful in current AML treatment if the text also included a further review of the significance for each treatment method.
In VEN/AZA therapy, it is proposed to suspend venetoclax in order to avoid prolongation of neutropenia if a bone marrow examination is performed in the middle and sufficient reduction of leukemia cells is obtained. I would like to hear your opinion on this.
We thank the reviewer for their comment. We have now added section 10: Day 14 Biopsy with Other AML Therapies on page 8. In this section we discuss the following:
- Day 14 Biopsy with Other AML Therapies
Most of the data available today for mid-induction bone marrow biopsy is in the context of ‘7+3” therapy. In the past 5-10y, we have seen several new drugs approved for the treatment of both newly-diagnosed or relapse/refractory AML. Here we discuss the mid-induction bone marrow biopsy data for the drugs approved for newly-diagnosed AML, acknowledging that is challenging to extrapolate the data from the ‘7+3’ population to this setting.
10.1 CPX-351
CPX-351, a liposomal formulation of cytarabine and daunorubicin, was approved for the treatment of patients with newly-diagnosed secondary AML based on a survival advantage compared to ‘7+3’ in a randomized trial. The trial focused on patients 60-75y with high-risk AML defined as either therapy-related, AML with antecedent MDS or CMML or AML with myelodysplasia related changes. The study administered a second induction for patients not achieving a hypoplastic D14 marrow. CPX-351 resulted in a higher CR rate compared to ‘7+3’ (37.3% vs. 25.6%, p=0.04). The composite remission rate was also higher with CPX-351 (55.2% vs. 34.0%) after one induction cycle. Interestingly, among patients with two induction cycles, there was no difference in response rate (CPX-351=31.3%, 7+3=35.3%), despite CPX-351 leading to a survival benefit for the entire population. Granular details of disease burden and timing of second induction were not provided to further analyze mid-induction biopsy results. Furthermore, the median time to neutrophil recovery was prolonged with CPX-351 with either one or two induction cycles (35days vs. 29 days). Given similar response rates to second induction as well as prolonged time to count recovery, the utility of a D14 biopsy in recipients of CPX-351 remains unclear and further studies are needed to understand it better.
10.2 HMA+VEN
The combination of HMA+VEN is now considered standard of care for older or unfit patients with newly-diagnosed AML. The initial study utilized 28 days of VEN. As HMA+VEN became routinely adapted in clinical practice, several modifications to the initial treatment schedule have been recommended to minimize the myelosuppressive toxicities of the regimen. One recommendation is to perform a biopsy between days 21 and 28 and to hold VEN if a morphological leukemia free state is achieved, as a measure to assist in count recovery. Several studies are now being conducted to optimize the dose and duration of VEN during induction therapy. More recently, the addition of a third agent to HMA+VEN (triplet therapy) is being investigated in clinical trials. Although these are not considered standard yet, the US Food and Drug Administration (FDA) has provided guidance in designing trial to manage the toxicities of the combination. Specifically, in the setting of triple therapy with HMA+VEN, a Day 14 bone marrow biopsy is recommended and in the setting of a morphological leukemia free state, VEN +/- the investigational agent can be held to assist in count recovery. Most of the mid-induction biopsy data in the setting of HMA+VEN is to guide holding further therapy instead of immediate re-induction. In the setting of disease on the biopsies performed in recipients of HMA+VEN, the recommendation is to continue VEN per the prescribing insert.
Since Table 1 does not include references, please add them. Also, please add information on how many patients with what cytogenetic risk were included, what regimen was used for treatment, and whether it was a prospective or retrospective study to Table 1.
We thank the reviewer for their comment. We have added the reference, cytogenetic risk, treatment details and retrospective/prospective information to Table 1.
Since there are multiple clinical trial protocols for treatment development that stipulate that re-induction therapy should be performed in the intermediate bone marrow examination, I think that most studies cannot be said to be retrospective studies. Please distinguish between developmental studies and retrospective studies.
We thank the reviewer for their comment. In Table 1 we have now mentioned which trials were retrospective studies and which were prospective. Furthermore, in the text we now describe which studies are clinical trials. Specifically, we state the prospective trials by Ofran et al. (ref 44), Kern et al. (ref 31), Bertoli et al. (ref 27), Erba et al. (ref 11) and Lancet et al. (ref 50) in the text while discussing them in various sections. We also discuss a prospective clinical trial that we are conducting to better understand the Day 14 Biopsy in AML (NCT06323590).
The same content is repeated throughout the text, so please organize the text.
We thank the reviewer for their comment. We have shortened the text and minimized repetition throughout the text.
References 44 and 45 are duplicated, so please correct them.
We thank the reviewer for their comment. We have deleted the duplicate reference.
Reviewer 3 Report
Comments and Suggestions for Authors The authors performed a review article on midinduction biopsies in patients with AML treated with "3+7" therapy as a prognosis on further treatment and outcomes.The article is comprehensive, gives the arguments that D14 biopsy has low specificity and sensitivity accompanied by the fact that there is no prospective data in the area. Furthermore, they highlight the fact that the significant subgroup of patients receive reinduction which may not be necessary. The article is clearly written, well organised and adequate "take-home" messages are given. I propose "accept without revision"Author Response
We appreciate the reviewer for their positive feedback.
Round 2
Reviewer 1 Report
Comments and Suggestions for Authors
Most comments requesting supplementation, specification, or reanalysis of the data were well addressed. The drawbacks I pointed out were also appropriately addressed.
Reviewer 2 Report
Comments and Suggestions for Authors
The authors appropriately added to the content of the paper in response to my comments, resulting in a more comprehensive and enriched review. This has made the paper even more valuable to readers.